# GATA4-targeted compounds induce apoptosis and diminish viability of hepatoblastoma cells

Sini M. Kinnunen[ID][1,2☉], Katja Eloranta[2,3☉], Marjut Pihlajoki[ID][2*], Mika J. Välimäki[1,4], Saana Pohjavaara[1], Emilie Indersie[ID][5], Stefano Cairo[5,6,7], Aarni Kuusinen[1], Heikki Ruskoaho[1‡], Markku Heikinheimo[2,8,9‡]

1 Drug Research Program and Division of Pharmacology and Pharmacotherapy, University of Helsinki, Helsinki, Finland, 2 Pediatric Research Center, Children's Hospital, University of Helsinki and Helsinki University Hospital, Helsinki, Finland, 3 Institute for Molecular Medicine Finland (FIMM), Helsinki Institute of Life Science (HiLIFE), University of Helsinki, Helsinki, Finland, 4 Cancer Cell Circuitry Laboratory, Translational Cancer Medicine, Medical Faculty, University of Helsinki, Helsinki, Finland, 5 Xentech, Evry, France, 6 Champions Oncology, Hackensack, New Jersey, United States of America, 7 Istituto di Ricerca Pediatrica, Padova, Italy, 8 Department of Pediatrics, Washington University School of Medicine, St. Louis, Missouri, United States of America, 9 Faculty of Medicine and Health Technology, Center for Child, Adolescent, and Maternal Health Research, Tampere University, Tampere, Finland

☉ equal contribution,
‡ equal contribution
* marjut.pihlajoki@helsinki.fi

## Abstract

Hepatoblastoma is the most common pediatric liver tumor, and it primarily affects young children. Despite improved treatment results due to modern chemotherapy and advanced surgical techniques, there is urgent need to improve the long-term prognosis of hepatoblastoma patients. Elevated expression of transcription factor GATA4 has been associated with pro-tumorigenic functions in hepatoblastoma. Herein, we explored the effects of a novel series of GATA4-targeted compounds (3i-2010–3i-2014) in several hepatoblastoma cell models. To this end, we assessed cell viability (2D and 3D), caspase 3/7 activity, cell cycle process with automated fluorescence microscopy, and RNA and protein expression using quantitative PCR, RNA-sequencing, and western blotting after treating cells with GATA4-targeted compounds. The GATA4-targeted compounds reduced hepatoblastoma cell viability but affected also healthy control cells when high doses were applied. Out of the five compounds used, the 3i-2012 was the most potent. This compound induced G0/G1 phase cell cycle arrest, increased caspase activity, and resulted in changes in transcriptome of hepatoblastoma cells. The differentially expressed genes (DEGs) were associated with cell cycle regulation. Transcription factor target analysis demonstrated that a notable proportion of DEGs were likely regulated by GATA4. In conclusion, 3i-2012 decreases hepatoblastoma cell survival by disturbing cell cycle regulation in the tumor cells.

**Data availability statement:** The RNA sequencing datasets generated and/or analyzed during the current study are available from the European Genome-phenome Archive (EGA) (https://ega-archive.org/, accession number EGAS50000000999/EGAD50000001463).

**Funding:** This work was supported by Jane and Aatos Erkko Foundation (HR; https://jaes.fi), Sigrid Jusélius Foundation (HR, MH; https://www.sigridjuselius.fi/en/), Helsinki University Hospital Research Funds (Helsingin ja Uudenmaan Sairaanhoitopiiri) (MH; https://www.hus.fi/en), Academy of Finland (Terveyden Tutkimuksen Toimikunta, Project 2666621) (HR; https://www.aka.fi/en/), Finnish Foundation for Cardiovascular Research (Sydäntutkimussäätiö) (HR; https://www.sydantutkimussaatio.fi/en/foundation). Sponsors or funders play no role in the study design, data collection and analysis, decision to publish, or preparation of the manuscript.

**Competing interests:** I have read the journal's policy and the authors of this manuscript have the following competing interests: SMK, MJV and HR are inventors in a patent application "Pharmaceutical compound" (PCT/FI2017/050661) concerning the compound 3i-1000 and its derivatives. EI is currently employed by the company XenTech. SC has formerly been employed by the company XenTech and is currently employed by the company Champions Oncology. No other competing interests to disclose. This does not alter our adherence to PLOS ONE policies on sharing data and materials.

## Introduction

Hepatoblastoma is the most common pediatric liver tumor primarily affecting young children. The median age of diagnosis is 18 months, and nearly all hepatoblastomas occur at any age of less than 5 years. The annual incidence of hepatoblastoma is 2.16 per million person-years [1]. The overall 5-year survival rate for hepatoblastoma is 77%, although this varies depending on the stage of the disease and the child's age [2]. Surgical removal of tumor is the curative treatment for hepatoblastoma. However, primary resection is recommended only for tumors that are categorized as very low risk [3]. Therefore, pre- and post-operative chemotherapy is necessary, and liver transplant may be needed in selected cases. The standard chemotherapy entails cisplatin, carboplatin, and doxorubicin. Additionally, irinotecan, 5-fluorouracil, vincristine, and etoposide can also be included in the treatment of advanced hepatoblastomas [4,5]. Despite aggressive chemotherapy, 25%–30% of initially unresectable tumors remain resistant to treatment and thus new options are urgently needed [6].

GATA family of transcription factors (GATA1–6) are double zinc finger proteins that are expressed early in embryonic development and control the diverse tissue differentiation [7–10]. GATA factors are involved in undifferentiated progenitor cell expansion, direction of coordinated maturation and cell cycle withdrawal in terminally differentiating cells [9]. Thus, alterations of GATA factors may contribute to the development of cancer. GATA4 is expressed in human fetal hepatocytes at gestational week 8 but its expression declines by gestational week 12 [11]. GATA4 functions as a pioneer transcription factor, which can bind in a closed chromatin state, increase the accessibility for other factors and thus promote liver-specific gene expression [12]. Interestingly, GATA4 is highly expressed in both pediatric hepatoblastomas and hepatocellular carcinoma (HCC) but absent in adult HCC [13]. Furthermore, GATA4 silencing by siRNA sensitizes HUH6 hepatoblastoma cells to the apoptotic effect of doxorubicin [14]. High GATA4 expression in hepatoblastoma cells favors a motile, mesenchymal-like phenotype and promotes proliferation [15,16].

Previously, we identified GATA4-targeted compounds (3i-1000 and 3i-2000) that inhibit the interaction with its co-factor NKX2–5 [17–19]. Additionally, our recent affinity chromatography and cellular thermal shift assay (CETSA) experiments provided evidence of direct physical binding of compounds selectively to GATA4 [19]. The most promising compound (3i-1000) is cardioprotective *in vitro* and *in vivo* [20-21] and modulates cardiac cell differentiation [20]. In the present study, we have designed a new family of GATA4-targeted compounds (3i-2011, 3i-2012, and 3i-2013) which modify GATA4 interactions with DNA and its co-factors more potently than the first generation of GATA4-targeted compounds. Given that GATA4 is known to regulate tumor-promoting functions in hepatoblastoma cells, we herein screened the effect of GATA4-targeted compounds on hepatoblastoma cell viability and survival.

## Results

### Identification of GATA4-targeted compounds and optimization of the lead

We have previously screened GATA4-targeted compounds and identified structurally distinct small molecules 3i-1000 and 3i-2000 (**S1 Fig**) [17,18]. To elucidate the effects

of novel GATA4-targeted compounds, we initially explored whether the structural modifications of compound 3i-2000 enhance the inhibitory effect against GATA4 and GATA4-NKX2–5 synergistic gene activation in luciferase assays (Fig 1a). All compounds were screened in COS-1 cells using the luciferase construct containing binding sites only for GATA4 and the construct containing binding sites only for NKX2–5. Of the new compounds, 3i-2011, 3i-2012, and 3i-2013 inhibited GATA4 reporter activity (Fig 1a) and synergistic GATA4-NKX2–5 gene activation (Fig 1b) more than the original lead compound 3i-1000 at 10 µM concentration. On the contrary, the structurally more-distant compounds 3i-2010 and 3i-2014 were less active up to 10 µM than the other compound analogues against the GATA4 promoter activity (Fig 1a).

The toxicity of the compounds was tested in fibroblast-like COS-1 cells and human induced pluripotent stem cells (hiP-SCs) using 3-(4,5-dimethylthiazol-2-yl)-2,5-diphenyl-tetrazolium bromide (MTT) test (Fig 1c,1d). The compound 3i-2012 at the concentration of 30 µM after 24-hour exposure was toxic to COS-1 cells. Other tested compounds did not markedly affect COS-1 cell viability (Fig 1c). In a previous study, hiPSCs were shown to be sensitive to compounds of 3i-1000 series [21]. The compounds of 3i-2000 series demonstrated similar diminishing effect on hiPSCs viability with concentrations of 3 µM and above, except 3i-2010 (Fig 1d).

To investigate potential other actions of 3i-2010 compounds, a commercial KINOMEscan screening and bromodomain panels were utilized. All tested 3i-2010 compounds inhibited kinase activity (>50%) of fms related receptor tyrosine kinase 3 (FLT3), KIT proto-oncogene, receptor tyrosine kinase (KIT), colony stimulating factor 1 receptor (CSF1R), platelet derived growth factor receptor alpha and beta (PDGFRA, PDGFRB), aurora kinase B (AURKB), and tropomyosin receptor kinase A (TRKA) (S1 Table, S2 Fig). Furthermore, a commercial bromodomain assay was employed to assess the impact of 3i-2012 on bromodomain protein inhibition. All the bromodomain proteins remained intact (S2 Table).

## The effect of GATA4-targeted compounds on hepatoblastoma cell viability

To study the effect of the compounds on hepatoblastoma cell viability, we first tested a novel 3i-2010 series of compounds and selected previously established compounds from 3i-1000 series in 2–4 hepatoblastoma cell models. The compounds of 3i-1000 series (3i-1000, 3i-1022, and 3i-1180) had no effect on hepatoblastoma cell viability at 24 hours but decreased survival approximately 20–50% after 48 hours exposure in HUH6 and HB-282 cells (S3 Fig). In response to 24 hours of treatment, most of the compounds in the 3i-2010 series (from 3i-2011 to 3i-2014) decreased hepatoblastoma cell viability at the highest concentration (30 µM) (Fig 2a–2g). The most prominent effect was observed with 3i-2012 which remarkably reduced cell viability already at a concentration of 10 µM. Similar decrease in HUH6 and HB-282 cell survival was observed after 48 hours treatment with 3i-2011, 3i-2012, 3i-2013, and 3i-2014 but this was evident already with smaller concentrations (≤10 µM) (Fig 2a–2g). As both HUH6 and HB-282 primarily represent embryonal histology, we also assessed the effect of 3i-2012 in hepatoblastoma cell lines of fetal origin (HB-295 and HB-303). Both lines showed reduced viability at the concentration of 3 µM after 24 hours of treatment, and in HB-303 cells viability was already markedly decreased at the concentration of 1 µM after 48 hours (Fig 2f–2g).

To validate these findings, we explored the effect of the most efficient 3i-2010 series compounds in two additional hepatoblastoma cell lines (Fig 2e,2g). PDX-derived HB-243 cells were especially sensitive to 3i-2012 and treatment with it reduced cell viability with the low concentration (1 µM) (Fig 2g). The second additional hepatoblastoma cell model HB-279 was more resistant to the compounds than other tested cell lines (Fig 2e,2g). To assess toxicity in the context of liver tissue, human primary hepatocytes were treated with the compound 3i-2012 which showed the greatest efficacy in hepatoblastoma cells. The viability of primary hepatocytes declined with concentrations above 3 µM both in response to 24 hours as well as 48 hours treatment (Fig 2f–2g).

## The impact of 3i-2011 and 3i-2012 on GATA4 expression in hepatoblastoma cells

To explore whether the newly discovered compounds affected GATA4 levels, we analyzed the RNA and protein expression in hepatoblastoma cells. Since 3i-2011 and 3i-2012 showed the most promising efficacy, these compounds were selected

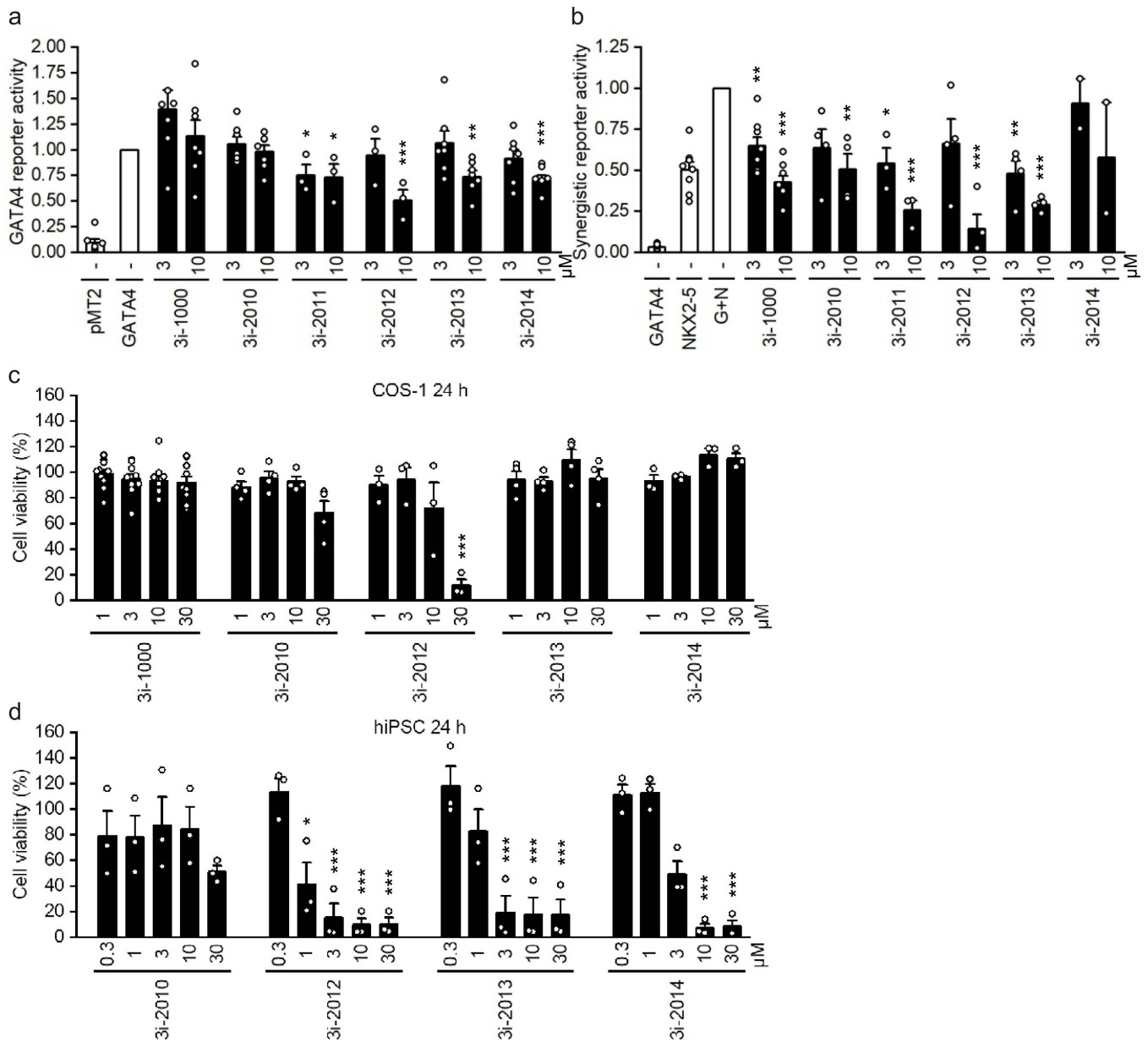

**Fig 1. The effect of small molecules on GATA4 transactivation and GATA4-NKX2-5 synergistic transactivation and cell toxicity.** COS-1 cells were transfected for 6 hours after which the small molecules were added in culture media for 24 hours. **(a)** The cells transfected with GATA4 binding sites containing reporter plasmid together with GATA4 expression plasmid or **(b)** three NKX2-5 binding sites containing reporter together with both GATA4 and NKX2-5 expression plasmids (G+N). Empty plasmid (pMT2) was used as control. The cell toxicity of GATA4 targeting compounds was assayed by MTT assay in COS-1 (c) and hiPSC (d) cells after 24 h incubation. The Fig shows the average, +SEM and the result of each independent experiments, n ≥ 3, except for 3i-2014 at panel B n = 2. * p < 0.05, ** p < 0.01, *** p < 0.001 vs. **(a)** GATA4 or **(b)** G+N (one-way ANOVA followed by the Tukey test or Welch ANOVA followed by the Games-Howell test).

for further assessments. The GATA4 mRNA and protein expression remained unaltered in response to 48 h of treatment with 3i-2011 or 3i-2012 (**S4 Fig**) in line with the results showing that the compounds act by inhibiting GATA interactions with its co-factors and binding directly to GATA4 [17-19]. Of note, due to technical reasons, the concentrations of the compounds used in GATA4 expression experiments were lower than those which decreased hepatoblastoma cell viability.

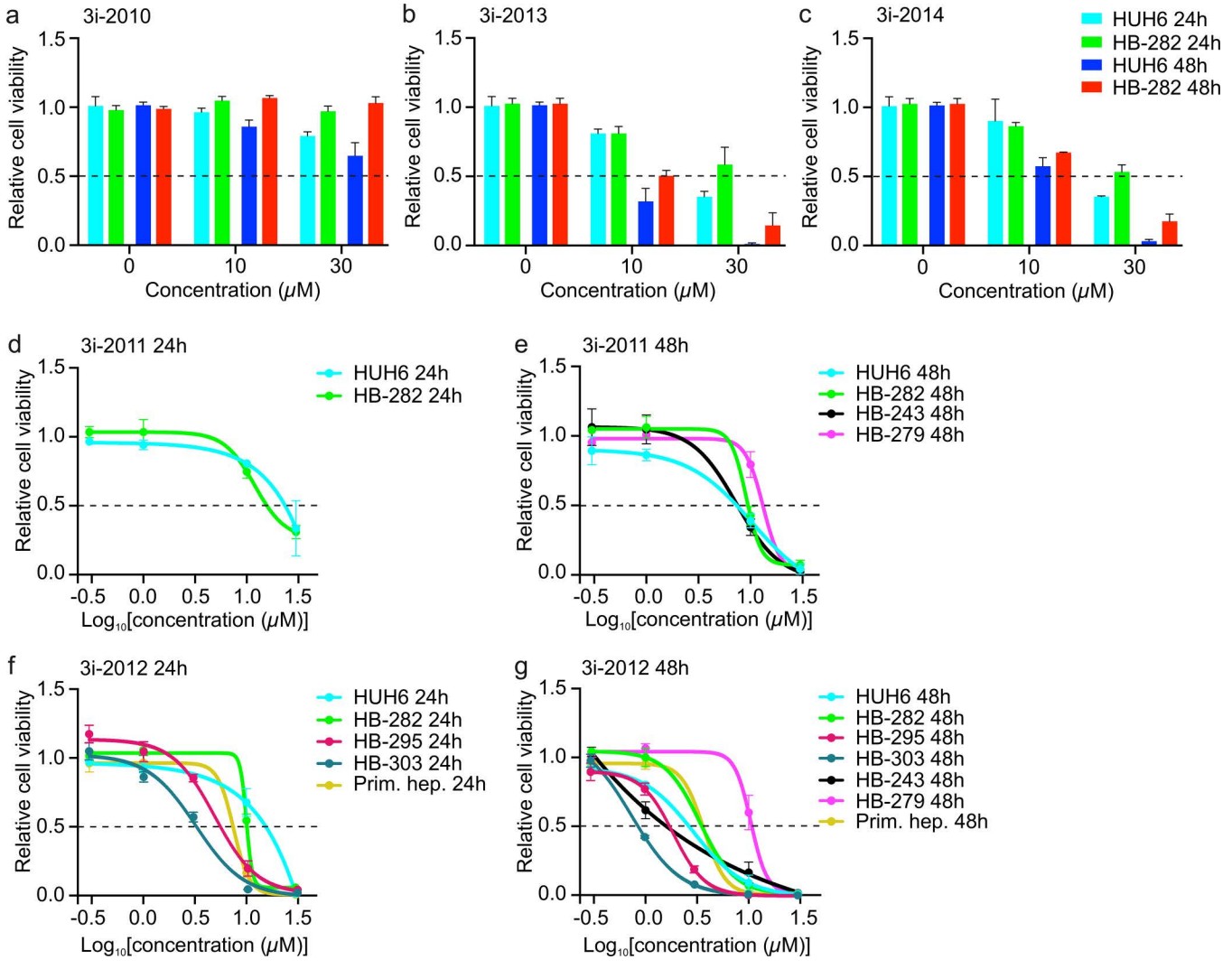

**Fig 2. The effect of small molecules on hepatoblastoma cell viability.** Relative ATP concentration in HUH6 and HB-282 cells after 24 and 48 h treatment with 3i-2010 **(a)**, 3i-2013 **(b)**, and 3i-2014 **(c)**. Relative ATP concentration in HUH6, HB-282, HB-303, HB-295, HB-243, and HB-279 cells after 24 and 48 h treatment with 3i-2011 (d and e) and 3i-2012 **(f and g)**. Results are presented as relative values of mean ± SD (N = 3). Dashed line indicates 50% cell viability.

## GATA4-targeted compounds' influence on hepatoblastoma spheroid viability

As spheroids more closely resemble the 3-dimensional architecture of solid tumors, the effect of compounds 3i-2011 and 3i-2012 were examined also on hepatoblastoma cell spheroid viability. HB-243 spheroids were let to grow 72 hours and HUH6 cells 48 hours before adding the compounds for 48 hours. The spheroids were imaged before adding the compound (0 hours) and after 48 hours of incubation, and the amount of ATP was quantified from the same samples. Extended dark core and loss of defined edges are characteristics of dying spheroids [22]. These morphological changes were observed in hepatoblastoma spheroids treated with 3i-2012 (**Fig 3a**,**3b**). Spheroids treated with 3i-2011 were more intact than cultures treated with 3i-2012, and only modest morphological changes were evident with the highest concentration (3 µM) (**Fig 3a**,**3b**). The quantification of ATP in spheroids showed that 3i-2012 at a

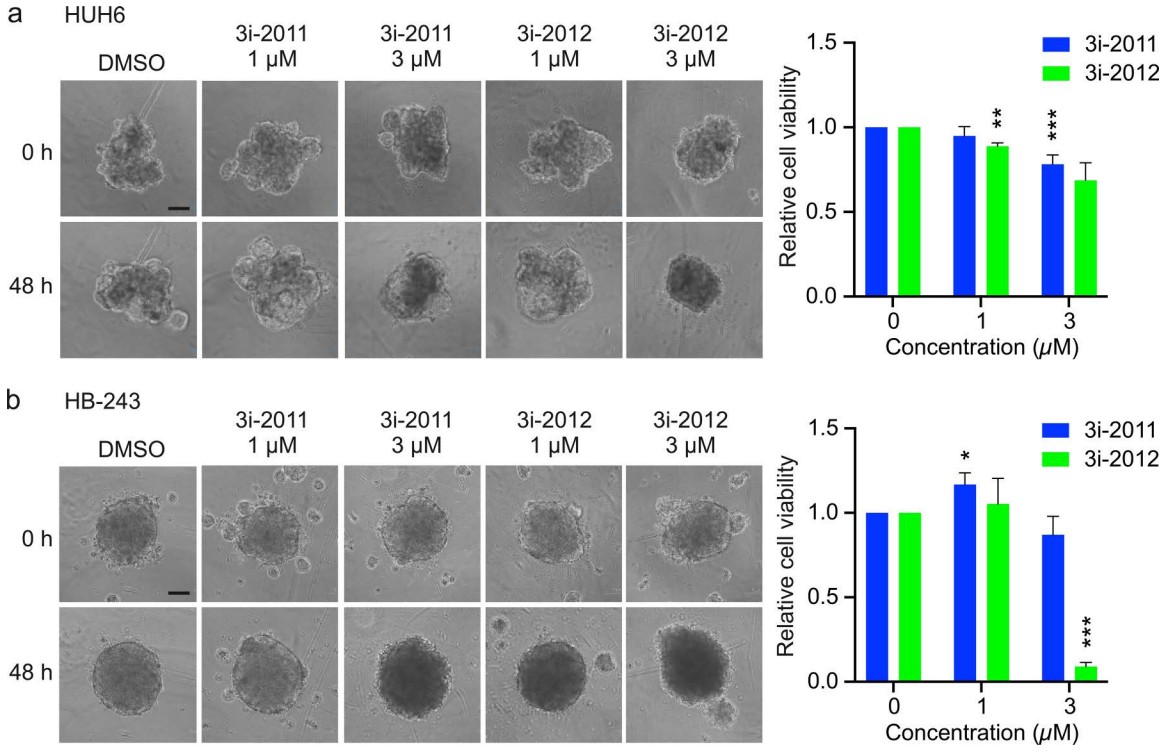

**Fig 3. The effect of small molecules on HB-243 and HUH6 spheroid viability.** Representative images and quantification of HUH6 **(a)** and HB-243 **(b)** spheroids. Quantification of spheroid viability after 48-hours was determined by measuring amount of ATP. * p < 0.05, ** p < 0.01, *** p < 0.001 vs. DMSO (one-way ANOVA followed by the Tukey test or Welch ANOVA followed by the Games-Howell test). Scale bar = 10 μm.

concentration of 3 μM decreased HB-243 spheroid viability by 85% while HUH6 cell viability did not reduce more than 25% (**Fig 3a**,**3b**). Treatment with smaller concentration of 3i-2012 (1 μM) caused only a mild alteration in cell viability. Also, the effect of 3i-2011 on hepatoblastoma spheroids was negligible and partly inconsistent between the cell lines (**Fig 3a**,**3b**).

## Caspase activity analysis and nuclear intensity in hepatoblastoma cells

To further assess the mechanism of action of GATA4-targeted compounds we then analyzed caspase activity by high-content analysis (HCA) in HUH6 cells. The treatment with compound 3i-2012 at 1 μM concentration for 24 hours did not significantly increase caspase-3/7 activity compared to DMSO treated cells (**Fig 4**). However, the 10 μM concentration significantly increased the number of caspase 3/7 positive cells (**Fig 4a**,**4b**).

HCA was also utilized to analyze the cell division cycle by quantifying the Hoechst staining intensity in nuclei. Treatment with volasertib was used as a positive control (**Fig 4b** and **4f, and S5d Fig**) [23]. The representative histogram shows a strong peak for G0/G1 phase at $1.25 \times 10^5$ in DMSO-treated cells (**Fig 4c**). The second peak for G2/M phase at $2.25 \times 10^5$ is much lower suggesting that cell division phase is fast in HUH6 cells. The histogram after treatment with 1 μM concentration of 3i-2012 is comparable to that of DMSO, showing no effects on cell cycle at this concentration (**Fig 4d**). Following treatment with 10 μM of 3i-2012 a clear second peak at $2.25 \times 10^5$ was noted, indicating an increase in the number of cells in G2/M phase, yet G0/G1 peak being still prominent (**Fig 4e**). Note that the number of the nuclei analyzed is lower due to cell death. Histograms of all the experiments are presented in **S5 Fig**

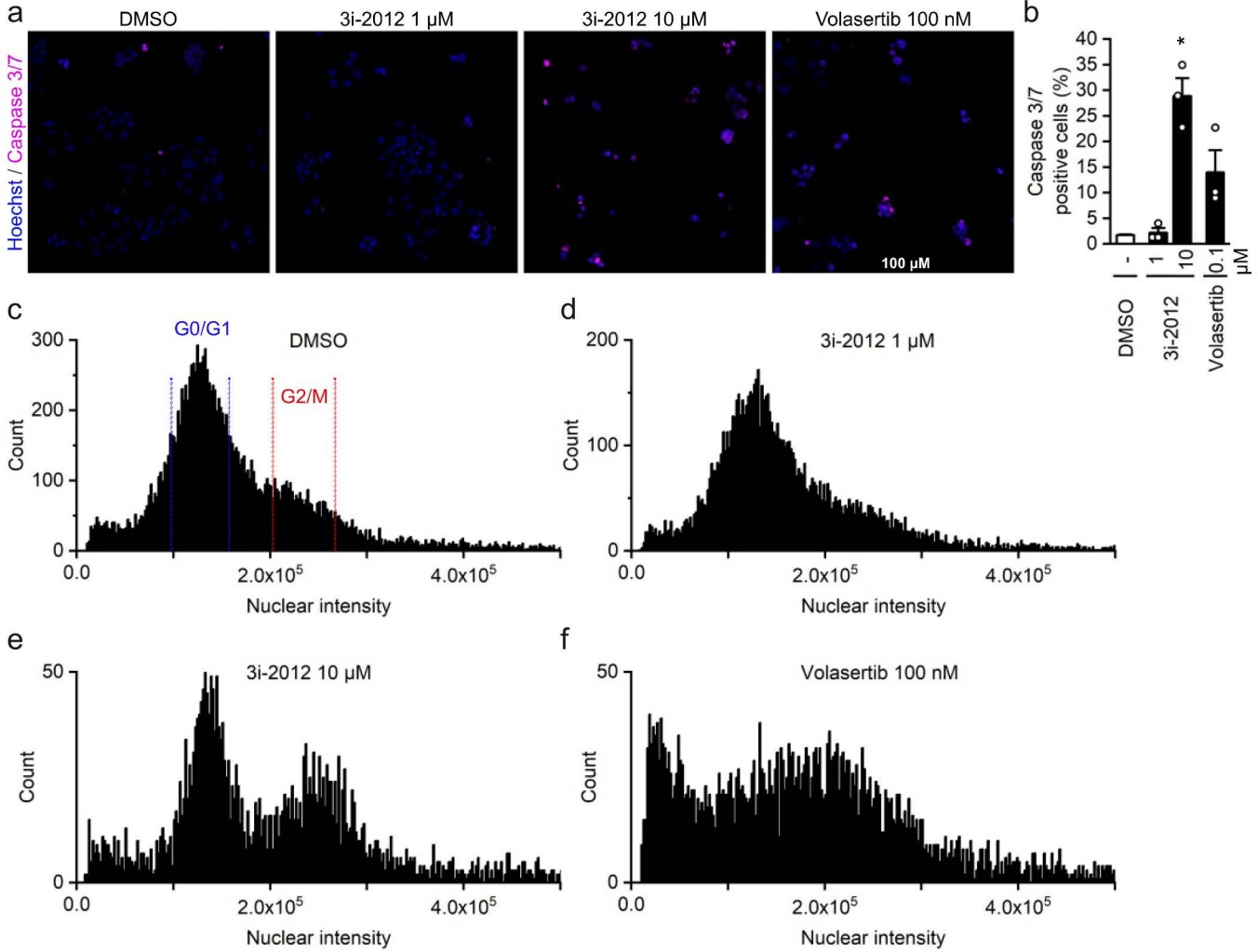

**Fig 4. Caspase 3/7 staining and nuclear intensity after 24-hours compound exposure in HUH6 cells.** Representative images of caspase 3/7 and Hoechst staining **(a)**. HCA was used to identify and quantify the caspase 3/7 positive cells **(b)**. The Fig shows the average, +SEM and the result of each independent experiments n = 3. Representative histograms of nuclear intensity analysis by HCA **(c-f)**. Histograms of all the experiments analyzed are presented in **S5 Fig** * p < 0.05 vs. DMSO (Welch ANOVA followed by the Games-Howell test). Volasertib served as a reference compound.

## Gene expression alterations in hepatoblastoma cells treated with 3i-2012

Finally, to examine the effects of the compounds on hepatoblastoma cell transcriptome, we evaluated changes in gene expression by RNAseq in hepatoblastoma cells after treating the cells for 24 hours with 3i-2012 at the concentrations of 300 nM or 1 μM. HB-243 cells were chosen for these experiments. Gene expression profiles of hepatoblastoma cells treated with lower concentration of 3i-2012 were slightly affected, as only 8 genes were differentially expressed in HB-243 cells (**Fig 5a**). Treatment with 1 μM of 3i-2012 induced more prominent gene expression changes, resulting in upregulation of 1062 and downregulation of 968 genes (**Fig 5b**). Next, we explored the transcriptional changes of *GATA4*, *NKX2–5*, and potential alternative targets identified by KINOMEscan assay to scope the effect of 3i-2012 on their expression. In line with our qPCR analysis, *GATA4* gene expression was not notably affected by 3i-2012 treatment (**S6 Fig**). Of the potential alternative targets, only *PDGFRB* was differentially expressed (lg$_2$ FC 1.1) in HB-243 cells (**S6 Fig**) in response

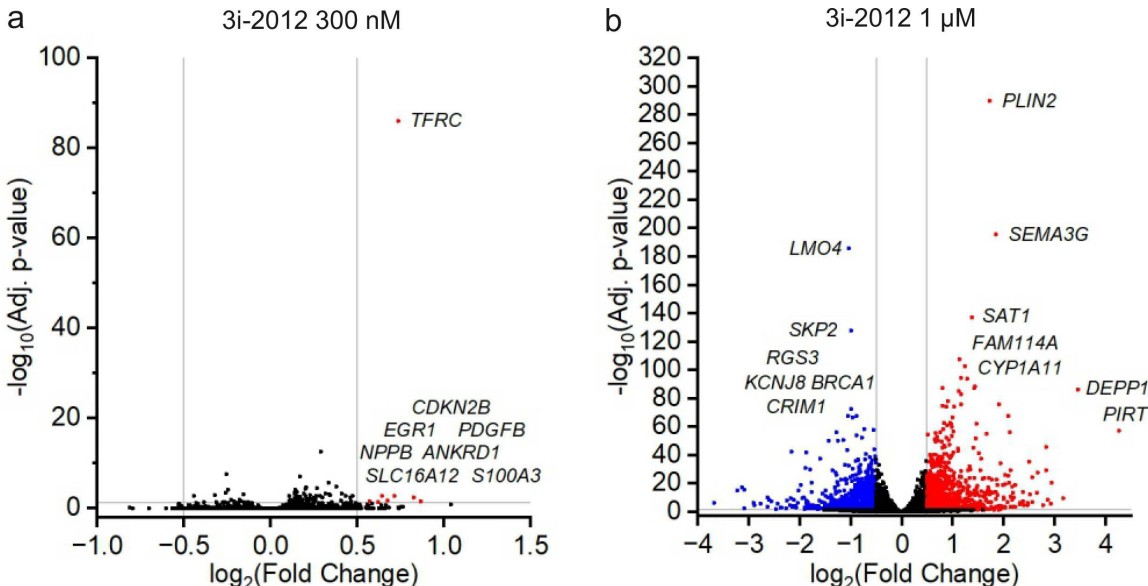

**Fig 5. Changes in gene expression evaluated by RNAseq after treating HB-243 cells for 24 hours with 300 nM and 1µM concentrations of 3i-2012.** Volcano blot analyses show the effect of compound on gene expression. **(a)** HB-243 cells exposed to 300 nM and **(b)** HB-243 cells exposed to 1 µM concentration of 3i-2012.

to the treatment with 3i-2012 (**S6 Fig**). The compound had minor effect on *AURKB* expression (**S6 Fig**) while *PDGFRA* remained unaltered (**S6 Fig**). Expression of *FLT3*, *KIT*, and *TRKA* was negligible both in control and 3i-2012 treated hepatoblastoma cells.

### Cell cycle related genes were enriched in hepatoblastoma cells treated with 3i-2012

The number of differentially expressed genes was very small in HB-243 hepatoblastoma cells treated with 300 nM of 3i-2012 and therefore this data was excluded from further gene set enrichment analyses. Enriched GO terms connected with biological processes, cellular components, and molecular functions were then analysed in HB-243 cell lines exposed to 1 µM of 3i-2012. Differentially expressed genes were associated with cell cycle process, especially the replication of DNA was emphasized (**Fig 6a**). Cellular component analysis highlighted structures relevant to basolateral plasma membrane (**Fig 6b**). The structures essential for DNA replication were also highlighted in cellular component analysis (**Fig 6b**). Consistent with the biological process and cellular component analysis, molecular functions linked to DNA replication were overrepresented in HB-243 cells (**Fig 6c**). Other enriched molecular functions were related to cellular transportation (**Fig 6c**).

In line with the enriched GO terms, KEGG and Reactome pathway analyses demonstrated a notable enrichment of cell cycle, DNA replication, and DNA repair system pathways in HB-243 cells treated with 1 µM of 3i-2012 (**Fig 7a**). Especially genes connected to G2/M checkpoints, biological oxidations, and G1/S transition were downregulated in Reactome pathway analysis (**Fig 7b**–**7d**). To investigate which transcription factors participate in the regulation of the differentially expressed genes, ChEA3 transcription enrichment analysis was carried out [24]. Interestingly, GATA4 was among the ten most enriched transcription factors in 3i-2012 treated cells (**Fig 7e**). Furthermore, SRY-Box Transcription Factor 13 (SOX13) and GATA4 were equally the most enriched transcription factors (**Fig 7e**).

a Biological Process

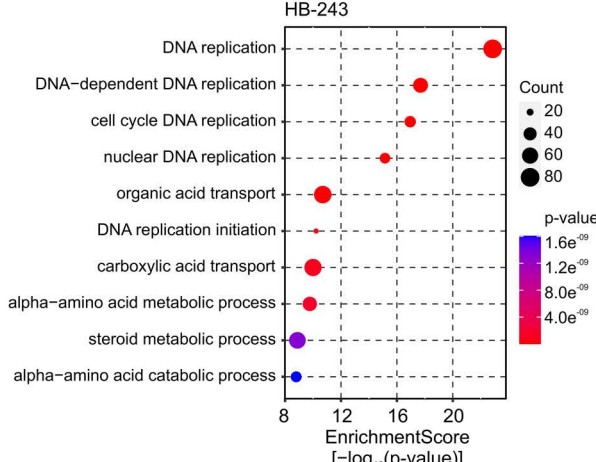

b Cellular Component

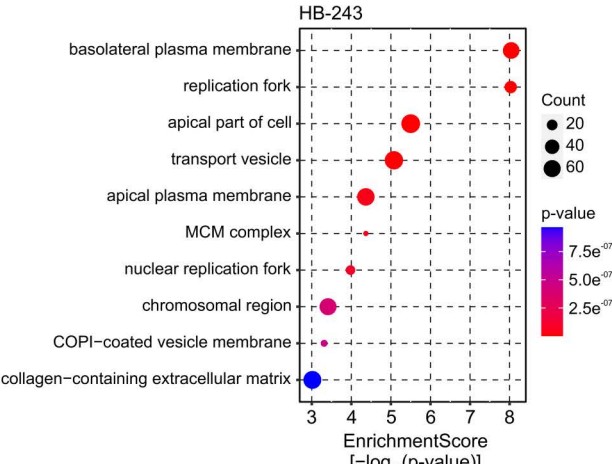

c Molecular Function

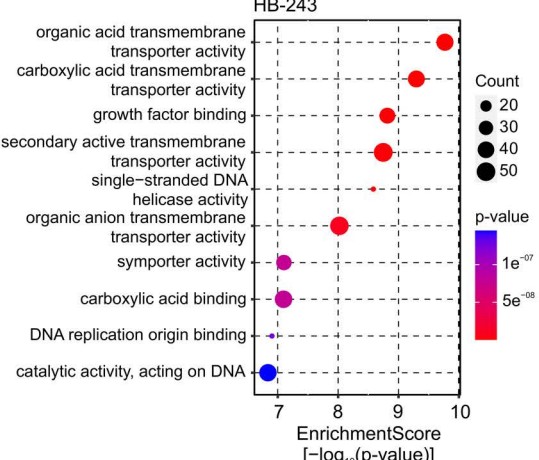

**Fig 6. Gene ontology enrichment analyses of hepatoblastoma cells exposed to 1 µM of 3i-2012 for 24 h.** Affected biological processes **(a)**, cellular components **(b)**, and molecular functions **(c)** in HB-243 hepatoblastoma cells treated with 3i-2012 are shown. All differentially expressed genes were used as an input.

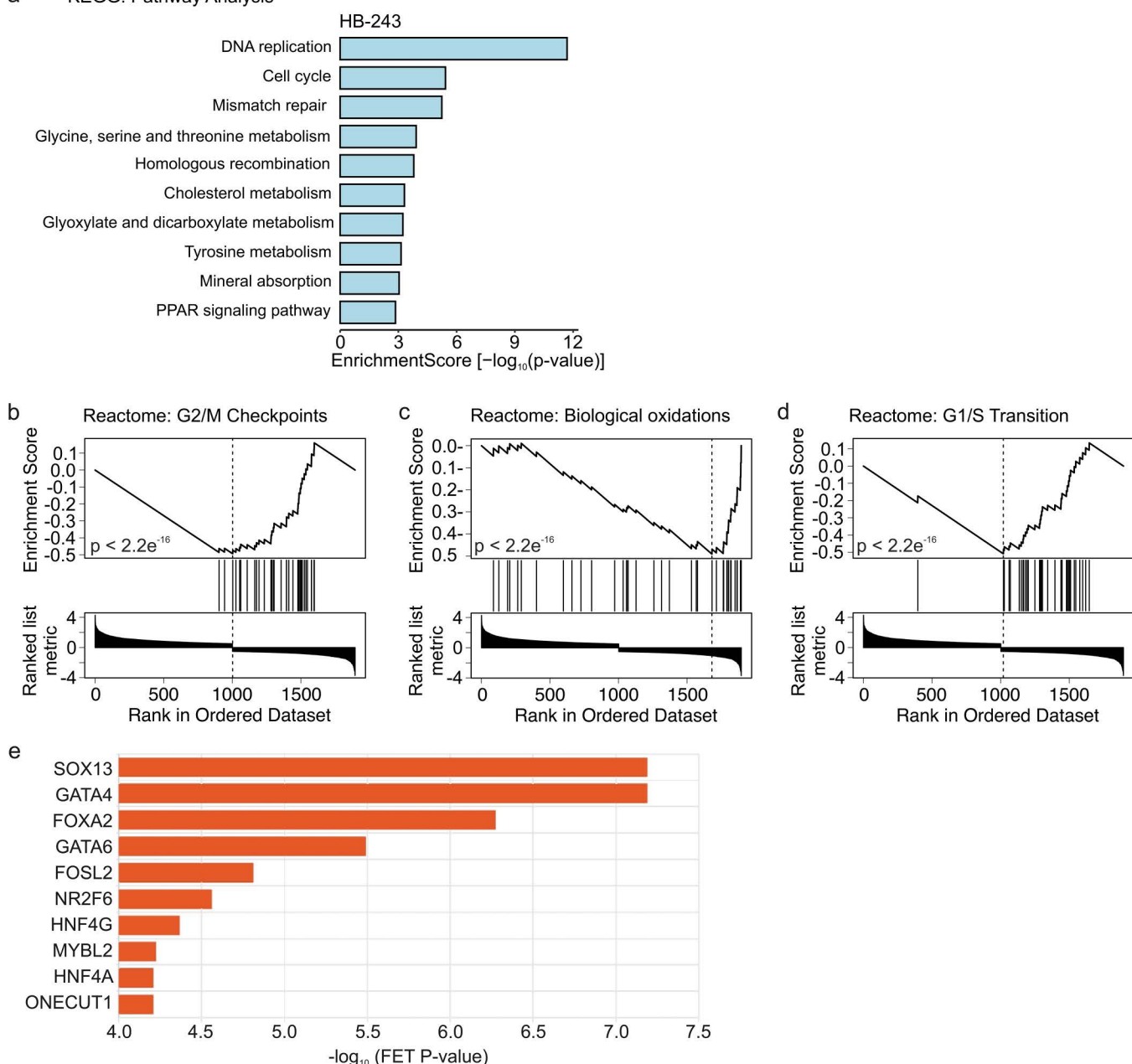

**Fig 7. KEGG and Reactome pathway analyses of HB-243 cells treated with 1µM of 3i-2012 for 24 h.** The ten most enriched pathways based on KEGG database in hepatoblastoma cells exposed to 3i-2012 **(a)**. Downregulation of G2/M checkpoints **(b)**, biological oxidation **(c)**, and G1/S transition (d) related genes were observed in HB-243 cells after 3i-2012 treatment. The three most enriched pathways are shown in GSEA plots **(b-d)**. Transcription factors which have potential to regulate the differentially expressed genes HB-234 cells **(e)**. The ten most enriched transcription factors in HB-243 gene set are shown in panel **e.**

## Discussion

Up to one-third of high-risk hepatoblastoma patients demonstrate an inadequate response to the current treatments [4,5]. Moreover, conventional chemotherapy compounds, such as doxorubicin and cisplatin, which are the standard-of-care in hepatoblastoma management, have severe effects also on healthy organs limiting their use. Transcription factors are a unique class of drug targets which provide multiple potential therapeutic approaches [25]. Here we explored the effects of previously described GATA4-targeted lead compound 3i-1000 and its novel structural derivatives, 3i-2000 family, on hepatoblastoma cells with elevated GATA4 expression [13].

The structural modifications of compound 3i-2000, especially morpholine-extensions, showed enhanced inhibitory effect on GATA4 reporter activity and interfered interaction with a GATA4 co-factor NKX2–5 resulting inhibition of synergistic gene activation in fibroblasts. The most potent inhibitor of GATA4 reporter activity and GATA4-NKX2–5 synergistic activity was the compound 3i-2012 which had also the highest inhibitory effect on the hepatoblastoma cell viability. On the other hand, the compound 3i-2010 did not affect GATA4 reporter activity and had only a weak effect on GATA4-NKX2 synergistic activity, and interestingly, also a minimal response to 3i-2010 treatment in the hepatoblastoma cells. Therefore, an association between the efficacy of the compounds to inhibit GATA4 or GATA4-NKX2 activity and their effect on hepatoblastoma cell survival appear to exist. However, small interfering RNA mediated silencing of *GATA4* in hepatoblastoma cells did not result in impaired cell viability or enhanced apoptosis in a previous study [14]. Thus, the observed effects of 3i-2000 series compounds on hepatoblastoma cell survival may be mediated by GATA4 co-operation with its interaction partners than solely GATA4. In addition, it is noteworthy that kinase activity assays demonstrated additional potential targets of tested drugs: FLT3, KIT, CSF1R, PDGFRA, PDGFRB, AURKB, and TRKA. Of these, *CSF1R*, *PDGFRB*, and *AURKB* mRNA expressions were affected by 3i-2012 exposure implying that these kinases might be related to mechanism of action of the compounds. A more complete target deconvolution, such as biochemical binding assays in hepatoblastoma cells or genome-wide CRISPR-based dependency analyses, would further strengthen the mechanistic insights and should be addressed in the future work. As published case reports and preclinical studies support the idea that multikinase inhibition may be a relevant strategy in hepatoblastoma management [26,27], it would be interesting to elucidate the role of kinases in hepatoblastoma cell response to 3i-2012 in upcoming studies.

Especially, the compound 3i-2012 remarkably reduced hepatoblastoma cell viability and enhanced caspase 3/7 activity at the concentrations 3 µM and above. The reduction in viability was evident both in 2-dimensional cell cultures and one of tested hepatoblastoma spheroid model. Hepatoblastoma is a histologically heterogeneous tumor, and the herein tested cell models also have slightly different main histological component [28,29]. The two best responders, HB-243 and HB-303, represent distinct histologies: HB-243 is predominantly embryonal, whereas HB-303 is of fetal origin. This finding suggest that the histological subtype may not influence the efficacy of these compounds. Although the treatment with 3i-2012 reduced hepatoblastoma cell survival, it also affected viability of healthy control hepatocytes and hiPSCs: the concentrations above 3 µM decreased the viability of primary hepatocytes more than 50% and hiPSCs were even more sensitive. Thus, the therapeutic window between efficacy and toxicity of 3i-2012 appears to be rather narrow and this should be considered in the future development of 3i-2000 series molecules.

To get insights to the effects of 3i-2012, we performed RNA-sequencing on HB-243 hepatoblastoma cells. Interestingly, the enriched gene sets were associated with cell cycle regulation. GATA factors are known to function either in undifferentiated progenitor cells and play a role in their expansion, or in terminally differentiating cells to coordinate maturation and cell cycle withdrawal [9]. Cyclin D2 and cyclin dependent kinase 4, essential for G1/S transition [30,31], have been shown to be direct targets of GATA4 in heart [32]. In line with the GATA4 role in other circumstances, genes related to G1/S transition as well as G2/M checkpoints were downregulated in HB-243 cells treated with 3i-2012. The nuclear intensity measurements support these results showing that most of nuclei are in G0/G1 after 10 µM 3i-2012 treatment compared to volasertib treated cells which are arrested at the G2/M phase. Furthermore, we observed reduced expression of genes participating in DNA replication process. The GATA4 co-factor NKX2–5 has a role in DNA repair on replication through

regulation of coiled-coil-domain-containing 117 (Ccdc117) [33]. Taken together, gene alterations observed in response to 3i-2012 exposure support the hypothesis that at least part of the actions of this compound are mediated via inhibition of GATA4 and its interaction partners. A notable proportion of differentially expressed genes were also putative GATA4 targets strengthening the concept that GATA4 is a relevant target for 3i-2012.

The hepatoblastoma cell models assessed in this study have different histologic and genetic profiles representing varying subtypes of hepatoblastoma [23]. Further investigations are needed to understand the exact mechanisms which generate hepatoblastoma cells vulnerable to 3i-2012 exposure and which biomarkers may predict the sensitivity to this treatment. Moreover, it remains unclear whether the selected compounds are effective *in vivo*. Thus, it is important to study the pharmacokinetics and efficacy the novel GATA-targeted compounds in animal models of hepatoblastoma.

## Conclusions

The 3i-2000 family of GATA4-targeted compounds reduced hepatoblastoma cell survival. The most potent small molecule 3i-2012 decreased hepatoblastoma cell viability and enhanced apoptosis. Treatment with 3i-2012 induced downregulation of genes relevant to cell cycle regulation and the mechanism of action of the compound may be related to these pathways. Furthermore, a notable proportion of differentially expressed genes in response to 3i-2012 is regulated by GATA4.

## Materials and methods

### Compounds

The synthesis of small molecules 3i-1000, 3i-1022, 3i-1180 and 3i-2000 have been described previously [17,18]. A novel series of compounds 3i-2010 – 3i-2014 were synthesized by Enamine (Kyiv, Ukraine). Volasertib was purchased from MedChemExpress (Monmouth Junction, NJ, USA). All compounds purchased were tested by HPLC−mass spectrometry and proton nuclear magnetic resonance by the vendor to confirm the sample identity and ensure a minimum purity of 95%.

### Kinase and bromodomain protein assays

The potential kinase and bromodomain activity of compounds in 3i-2010 series (1 μM or 10 μM) was analyzed with commercial KINOMEscan or BROMOscan assays (DiscoverX, Eurofins).

### Cell cultures

Fibroblast-like COS-1 cells were purchased from DSMZ (Braunschweig, Germany) and cultured according to manufacturer's instructions. The human induced pluripotent stem cells (hiPSC) (IMR-90)-4 line was purchased from WiCell (Madison, Wisconsin, USA). The cell culture and differentiation procedure has been described previously [34]. Immortalized human hepatoblastoma cell line HUH6 was purchased from Japanese Collection of Research Bioresources Cell Bank (Osaka, Japan) and cultured in low glucose DMEM (#21885, Gibco, Waltham, MA, USA) supplemented with 10% fetal bovine serum (FBS), 100 U/ml penicillin, and 100 μg/ml streptomycin. Previously established human patient derived xenograft (PDX) hepatoblastoma cell lines HB-243, HB-279, HB-282, HB-295, and HB-303 were obtained from XenTech (Evry, France) and the clinical information has been described earlier [28]. The cells were cultured in Advanced DMEM/F12 (#12634, Gibco) supplemented with 8% FBS, 2μM L-glutamine (#35050. Gibco), 100 U/ml penicillin, 100 μg/ml streptomycin, and 20 μM rock kinase inhibitor Y-27632 (SelleckChem). Cryopreserved human hepatocytes (HUM191441, Caucasian female age of 19 months) were purchased from Lonza (#HUCPG, Antwerp, Belgium). Hepatocyte thawing, plating and maintenance was performed according to manufacturer's instructions. Cultures were maintained at 37°C in a humified incubator containing 5% $CO_2$.

## Luciferase assay

The screening assays, GATA4 transactivation and GATA4-NKX2–5 synergy assays, have been described earlier [18,35]. Shortly, COS-1 cells were transfected for six hours with reporter plasmid NP-112 containing binding sites for GATA4 or pGL3-3xHA containing binding sites for NKX2–5 in addition to expression vectors for GATA4 and NKX2–5 proteins, pMT2-GATA4 and pMT2-NKX2–5, respectively. The plasmids have been described earlier [35–37]. The cells were treated with compounds or vehicle (0.05% DMSO) for 24 hours.

## Toxicity screening assays

The toxicity tests lactate dehydrogenase (LDH) and MTT assays were carried out in COS-1 cells and hiPSC as described previously [18,21].

## Cell viability assays

HUH6 cells were plated in density of 20,000 cells, HB-282 7,000 cells, HB-295 15,000 cells, HB-303 15,000 cells, HB-243 10,000 cells, and HB-279 15,000 cells per well on Isoplate-96 microplates (PerkinElmer, Turku, Finland) and grown over night. The cells were treated with compounds or vehicle (0.1% DMSO) for 24 h or 48 h. The viability was measured by quantifying the ATP using the luminescence-based CellTiter-Glo® -kit (Promega, Madison, WI, USA) or the ATPLite™ 2D monitoring system (PerkinElmer) with Glomax 96 microplate luminometer (Promega) or Victor2 plate reader (Perkin Elmer). To measure viability of spheroids, HUH6 and HB-243 cells were plated in density of 2,000 cells per well on CellCarrier spheroid ULA 96-well plates (PerkinElmer). Spheroids were let to establish for 48 (HUH6) or 72 hours (HB-243). After the establishment, the spheroids were treated with compounds or vehicle (0.05% DMSO) for 48 h. The viability was measured using Cell-Titer Glo as described above, however; the samples were transferred to OptiPlate-96 before reading the luminescence.

## Real time quantitative PCR (RT-qPCR)

HB-243 cells were plated on 6-well plates in densities of 300,000–350,000 cells per well. On the next day, the cells were treated with compounds or vehicle (0.1% DMSO) for 48 hours. RNA and protein were isolated using NucleoSpin RNA/Protein purification kit (#740933, Macherey-Nagel, Düren, Germany).

For RT-qPCR, 600 ng of total RNA was reverse transcribed in 10 µL reactions using an iScript cDNA synthesis Kit (#1708891, Bio-Rad, Hercules, CA, USA) according to the manufacturer's protocol using random hexamer primers and an SimpliAmp Thermal Cycler (Applied Biosystems by Life Technologies, Waltham, MA, US). The cDNA was diluted 1:12 in HyClone Molecular Biology-Grade Water (Cytiva, South Logan, Utah, USA) and stored at −80 °C.

The following primers were used with PowerUp SYBR Green Master Mix (#A25742, Applied Biosystems by Thermo Fisher Scientific) according to the manufacturer's instructions: GATA4 5'-ATCTAAGACACCAGCAGCTCCTTC-3' (forward), 5'-AGGCTCCGTCTTGATGGGAC-3' (reverse) amplifying 123 bp product spanning exons 5 and 6 of NM_002052; beta-2-microglobulin (B2M) 5'-GATGAGTATGCCTGCCGTGT-3' (forward), 5'-CTGCTTACATGTCTTGATCCCA-3' (reverse); peptidylprolyl isomerase G (PPIG) 5'-CAATGGCCAACAGAGGGAAG-3' (forward), 5'-CCAAAAACATGATGCCCA-3' (reverse). A CFX384 Real-Time System C1000 Touch Thermal Cycler (Bio-Rad) was used to analyze 5 µL of the cDNA dilution in 10 µL reactions on a white Hard-Shell 384-well PCR plates (HSP3805, Bio-Rad). Each reaction was run in triplicate, and the average of the technical replicates was used in the analysis as N = 1. The 2-ΔΔCt method was used to analyze the relative gene expression using geometric mean of B2M and PPIG expression as a reference.

## Western blot

Proteins were extracted using NucleoSpin RNA/Protein purification kit as described above. Ten micrograms of protein was separated by electrophoresis using Mini-Protean TGX Stain-Free Gels (#456–8126, Bio-Rad). Proteins were transferred

onto polyvinyl fluoride membrane using iBlot transfersystem (Invitrogen, Thermo Fisher Scientific) and non-specific binding was blocked with 5% non-fat milk in 0.1% TBS-Tween buffer (TBST). Membranes were incubated with anti-GATA4 antibody in 1:1000 dilution in 1% milk-TBST (sc-9053, Santa Cruz Biotechnology, Santa Cruz, CA, USA) at +4∘C for overnight and goat anti-rabbit IgG secondary antibody in 1:10,000 dilution in 5% milk-TBST (#711-035-152, Jackson ImmunoResearch, West Grove, PA, USA) at room temperature for 1 h. Protein bands were detected utilizing Enhanced Chemiluminescence detection kit (RPN2232, Amersham, GE Healthcare, Barrington, IL, USA) and analyzed with Image Lab Software 6.0 (Bio-rad). Band intensity was normalized to total protein amount in each lane using stain-free technology (see the S1 File_row_images) [38].

## Caspase activity and cell cycle analyses

To study caspase activation in HUH6 line, the cells were plated on 96-well Phenoplates (#6055302, Perkin Elmer), exposed to the compounds and just before 24 hour, incubated with 5 μM CellEvent™ Caspase-3/7 Green Detection Reagent solution (C10723, Thermo Fisher Scientific) in PBS with 5% FBS for 30 min and stained with 2.5 μg/ml Hoechst 33342 (#62249, Thermo Scientific Scientific) in PBS with 5% FBS for 10 min at 37 °C. Then the cells were fixed by adding 16% methanol-free paraformaldehyde (PFA, #43368, Thermo Fisher Scientific) in final 4% concentration for 20 min at room temperature and washed with PBS. For high-content analysis (HCA), the cells were imaged and analyzed with the CellInsight CX5 High-Content Screening Platform (Thermo Scientific) using a 10×objective (Olympus UPlanFL N 10×/0.3). Hoechst fluorescence was used to identify the cells and to define the nuclear area. The intensity of green fluorescence within the nucleus was used as a measure of caspase-3/7 activity. The threshold for caspase-3/7 positive and caspase-3/7 negative cells was set manually in each experiment. Increased intensity of Hoechst staining was used as a marker for DNA replication whereas decreased intensity was an indication of decrease in the amount of double-stranded DNA in the cells [39]. DNA intensity in nuclei was quantified with HCA. Each experiment included three technical replicates for compound treatments and five for DMSO control. Both analyses were performed from two images taken from each well.

## RNA-sequencing

HB-243 cells were plated on 6-well plates with a density of 350,000 cells per well and the following day treated with compounds for 24 hours. Total RNA was extracted using NucleoSpin RNA kit (#740955, Machery-Nagel). Further RNA quality analysis, RNA-sequencing and primary data analyses were performed by GENEWIZ (Leipzig, Germany). Briefly the workflow, Poly(A) selection was used for RNA selection method and samples were sequenced on Illumina NovaSeq 6000 yielding 2x150 bp paired end reads (GENEWIZ). Sequence reads were trimmed using Trimmomatic v.0.36. The trimmed reads were mapped to the GRCh38.p14 reference genome using the STAR aligner v.2.5.2b. Unique gene hit counts were calculated by using featureCounts from the Subread package v.1.5.2. Using DESeq2, a comparison of gene expression between the control group and compound treated group was performed. The Wald test was used to generate p-values and log2 fold changes. Genes with an Benjamini-Hochberg adjusted p-value < 0.05 and absolute log2 fold change above +0.5 or below −0.5 were called as differentially expressed genes for each comparison.

## Gene set enrichment analyses and data visualization

Gene Ontology (GO) enrichment and Kyoto Encyclopedia of Genes and Genomes (KEGG) pathway analyses and data visualizations were conducted with SRplot toolset [40]. Enrichment of Reactome pathways was assessed using Gene set enricment analysis (GSEA) tool in WebGestalt toolkits (version 2019, default settings, available at http://www.webgestalt.org/) [41]. ChEA3 transcription factor enrichment analysis was used to investigate which transcription factors participate in regulation of the differentially expressed genes [24]. For ChEA3 analyses, ReMap library was used as a reference.

## Statistics

All other statistical analyses were performed using IBM SPSS Statistics 25 software. Statistical significance was evaluated by one-way ANOVA followed by a Tukey post-hoc test. In case of unequal variances, Welch ANOVA was used followed by Games-Howell. A p-value of <0.05 was considered statistically significant. Curve fitting [log(agonist) vs. response – Variable slope (four parameters)] was performed with GraphPad Prism version 10.1.1 (GraphPad Software, Boston, Massachusetts USA, www.graphpad.com).

## Supporting information

**S1 File. The original, uncropped raw images of the western blots.**
(PDF)

**S1 Fig. Chemical structures of GATA4-acting compound 3i-1000 1 and 3i-2000 2 and their previously reported (3i-1020 2, 3i-1180 2) or novel compound analogues (3i-2010–2014) investigated in this study.** Based on the structural analysis, we introduced a variable morpholine-extensions to the 3i-2000 compound scaffold, leading to the discovery of novel GATA4-modulators 4-(tert-butyl)-N-(3-(4-(2-morpholinoethoxy)phenyl)-1H-pyrazol-5-yl)benzamide (3i-2011), 4-(tert-butyl)-N-(3-(4-(3-morpholinopropoxy)phenyl)-1H-pyrazol-5-yl)benzamide (3i-2012) and 4-(tert-butyl)-N-(3-(3-methyl-4-(2-morpholinoethoxy)phenyl)-1H-pyrazol-5-yl)benzamide (3i-2013) showing an improved inhibitory activity in luciferase assays in comparison to original lead compound 3i-1000.
(PDF)

**S2 Fig. TREEspot™ Interaction Maps of KINOMEscan™.** Kinases found to bind are marked with red circles, where larger circles indicate higher-affinity binding.
(PDF)

**S3 Fig. The effect of small molecules on HB cell viability.** Relative ATP concentration in HUH6 and HB-282 cells after 24 and 48 h treatment with 3i-2000 (a), 3i-2022 (b), and 3i-1180 (c). Results are presented as relative values of mean±SD (N=3). Dashed line indicates 50% cell viability.
(PDF)

**S4 Fig. Effect of small molecules on GATA4 mRNA (a) and protein levels (b-d) in HB-243 cells after 48 hours exposure.** The Fig shows the average,+SEM and the result of each independent experiments n≥3, except 3i-2011 3 µM n=2 and for 3i-2012 0.3 µM in panel A n=2. The original, uncropped raw images of the blots used to generate the western blot results are shown in S1 File.
(PDF)

**S5 Fig. High content analysis of the Hoechst staining intensity in nuclei after 24 hours exposure of DMSO (a), 1 µM 3i-2012 (b), 10 µM 3i-2012 (c), or volasertib (d) in HUH6 cells.** The histograms show the number of nuclei and Hoechst staining intensity quantified from three technical replicates.
(PDF)

**S6 Fig. The effect of 3i-2012 compound on *GATA4, NKX2–5*, and potential alternative targets identified by KINOMEscan assay on RNA expression in HB-243 cells.** The box represents the interquartile range, and the whiskers represent the 1st and 4th quartile. The line inside the box is the median.
(PDF)

**S1 Table. KINOMEscan™ results.** Selected kinases were screened for target validation by Eurofins DiscoverX (Fremont, CA, USA). The results are presented as % from control.
(PDF)

**S2 Table. BROMOscan™ results.** A panel of 32 bromodomain were screened for target validation by Eurofins DiscoverX (Fremont, CA, USA). The results are presented as % from control.
(PDF)

## Acknowledgments

We thank Ms. Liisa Lappalainen for technical assistance.

## Author contributions

**Conceptualization:** Sini M. Kinnunen, Marjut Pihlajoki, Mika J. Välimäki, Heikki Ruskoaho, Markku Heikinheimo.

**Data curation:** Sini M. Kinnunen, Katja Eloranta, Mika J. Välimäki.

**Formal analysis:** Sini M. Kinnunen, Katja Eloranta, Marjut Pihlajoki.

**Funding acquisition:** Heikki Ruskoaho, Markku Heikinheimo.

**Investigation:** Sini M. Kinnunen, Katja Eloranta, Marjut Pihlajoki, Mika J. Välimäki, Saana Pohjavaara, Aarni Kuusinen.

**Methodology:** Sini M. Kinnunen, Mika J. Välimäki.

**Resources:** Emilie Indersie, Stefano Cairo.

**Supervision:** Heikki Ruskoaho, Markku Heikinheimo.

**Validation:** Sini M. Kinnunen, Katja Eloranta.

**Visualization:** Marjut Pihlajoki.

**Writing – original draft:** Sini M. Kinnunen, Katja Eloranta.

**Writing – review & editing:** Sini M. Kinnunen, Katja Eloranta, Marjut Pihlajoki, Mika J. Välimäki, Saana Pohjavaara, Emilie Indersie, Stefano Cairo, Aarni Kuusinen, Heikki Ruskoaho, Markku Heikinheimo.

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
