## [Decision Letter · Decision Letter 0]

21 Nov 2025

Dear Dr. Pihlajoki,

Thank you for submitting your manuscript to PLOS ONE. After careful consideration, we feel that it has merit but does not fully meet PLOS ONE’s publication criteria as it currently stands. Therefore, we invite you to submit a revised version of the manuscript that addresses the points raised during the review process.

We look forward to receiving your revised manuscript.

Kind regards,

Consolato M. Sergi

Academic Editor

PLOS ONE

Journal Requirements:

[This work was supported by Jane and Aatos Erkko Foundation (HR; https://jaes.fi), Sigrid Jusélius Foundation (HR, MH; https://www.sigridjuselius.fi/en/), Helsinki University Hospital Research Funds (MH; https://www.hus.fi/en), Academy of Finland (Project 2666621) (HR; https://www.aka.fi/en/), Finnish Foundation for Cardiovascular Research (HR; https://www.sydantutkimussaatio.fi/en/foundation). Sponsors or funders play no role in the study design, data collection and analysis, decision to publish, or preparation of the manuscript.].

3. Thank you for stating the following in your manuscript:

[This work was supported by Jane and Aatos Erkko Foundation (HR; https://jaes.fi), Sigrid Jusélius Foundation (HR, MH; https://www.sigridjuselius.fi/en/), Helsinki University Hospital Research Funds (MH; https://www.hus.fi/en), Academy of Finland (Project 2666621) (HR; https://www.aka.fi/en/), Finnish Foundation for Cardiovascular Research (HR; https://www.sydantutkimussaatio.fi/en/foundation). Sponsors or funders play no role in the study design, data collection and analysis, decision to publish, or preparation of the manuscript.]

[This work was supported by Jane and Aatos Erkko Foundation (HR; https://jaes.fi), Sigrid Jusélius Foundation (HR, MH; https://www.sigridjuselius.fi/en/), Helsinki University Hospital Research Funds (MH; https://www.hus.fi/en), Academy of Finland (Project 2666621) (HR; https://www.aka.fi/en/), Finnish Foundation for Cardiovascular Research (HR; https://www.sydantutkimussaatio.fi/en/foundation). Sponsors or funders play no role in the study design, data collection and analysis, decision to publish, or preparation of the manuscript.]

[I have read the journal's policy and the authors of this manuscript have the following competing interests: SMK, MJV and HR are inventors in a patent application “Pharmaceutical compound” (PCT/FI2017/050661) concerning the compound 3i-1000 and its derivatives. EI is currently employed by the company XenTech. SC has formerly been employed by the company XenTech and is currently employed by the company Champions Oncology. No other competing interests to disclose.].

We note that one or more of the authors are employed by a commercial company: XenTech, Champions Oncology.

7. Please include captions for your Supporting Information files at the end of your manuscript, and update any in-text citations to match accordingly. Please see our Supporting Information guidelines for more information: http://journals.plos.org/plosone/s/supporting-information .

8. Please upload a new copy of Figure S2 and S5 as the details are not clear. Please follow the link for more information:  https://journals.plos.org/plosone/s/figures

Additional Editor Comments:

Please address all comments thoroughly.

Reviewers' comments:

Reviewer's Responses to Questions

**Comments to the Author**

1. Is the manuscript technically sound, and do the data support the conclusions?

Reviewer #1: Yes

Reviewer #2: Yes

2. Has the statistical analysis been performed appropriately and rigorously?

Reviewer #1: Yes

Reviewer #2: Yes

3. Have the authors made all data underlying the findings in their manuscript fully available?

Reviewer #1: Yes

Reviewer #2: No

4. Is the manuscript presented in an intelligible fashion and written in standard English?

Reviewer #1: Yes

Reviewer #2: Yes

Reviewer #1: Very interesting in-depth experiment on a novel group of GATA4 inhibitors. Comparison of the effectiveness of several compounds reinforces the utility of the most potent variant.

The prognosis of the hepatoblastoma cell line used in the experiment [embryonal] represents ~20% of all hepatoblastomas, with ~60% survival at 24 months. It would be of therapeutic interest to study the effectiveness of these inhibitors on hepatoblastomas with some of the other histologic types, e.g. pure fetal [~30% of all hepatoblastomas with a 24-month survival of ~90%], or small cell undifferentiated [~3% of all hepatoblastomas with a 24-month survival of 0%].

Excellent graphs and pictures.

Some typos:

Line 47: ... effects of a novel ...;

Line 61: ... occur at any age of ...;

Line 85: 'CETSA", not "CESTA".

Line 134: ... tested a novel ...;

Line 139: ... compounds in the 3i-2010 ...;

Line 140: ... viability at the highest ...;

Line 141: ... already at a concentration ...;

Line 169: ... compounds' influence ...;

Line 179: ... at a concentration ...;

Line 179: ... spheroid viability by 85% ...;

Line 244: ... Consistent with the ...;

Line 289: ... survival may be mediated ...;

Line 296: ... to 3i-2012 in upcoming studies ...;

Line 315: ... are arrested at the G2/M phase.;

Line 319: ... at least part of the actions ...;

Line 322: ... have different histologic and genetic ...;

Line 330: Furthermore, a notable ...;

Line 331: ... response to 3i-2012 is regulated ... [proportion ... is ...];

Line 334: The synthesis of ...;

Line 452: ... were performed using IBM ...;

Line 462: ... by the Jane ...; ... the Sigfrid ...;

Line 463: ... the Helsinki ....

Reviewer #2: In the manuscript entitled "GATA4-targeted compounds induce apoptosis and diminish viability of hepatoblastoma cells," the authors screened compounds targeting GATA4 to provide potential new drug candidates for treating diseases such as hepatoblastoma. This work is valuable; however, the following issues should be addressed before the manuscript can be considered for further processing.

The authors performed initial screening and identified compounds exhibiting cytotoxicity. Potential related mechanisms were investigated via high-throughput screening. Nevertheless, it remains unclear whether the selected compounds are effective in vivo. Additionally, the direct target proteins of these compounds have not been explored. This study would be significantly strengthened if the pharmacological and efficacy investigations were conducted in greater depth and supported by more comprehensive evidence.

**Do you want your identity to be public for this peer review?** For information about this choice, including consent withdrawal, please see our Privacy Policy

Reviewer #1: No

Reviewer #2: No

---

## [Author Response · Author response to Decision Letter 1]

17 Dec 2025

We thank the editor and the referees for their thorough and constructive critiques. Our point-by-point responses to the comments are given below. Text changes are marked as track changes in the body of the manuscript. We believe that the manuscript has been considerably improved by the changes made and hope that it now fulfills the criteria for publication in the Journal.

Editorial comments:

Response: The data is now fully available in the The European Genome-phenome Archive (EGA) database. This information has now been incorporated into the revised manuscript.

Response: Captions for the Supporting Information have now been added into the revised manuscript.

8. Please upload a new copy of Figure S2 and S5 as the details are not clear. Please follow the link for more information: https://journals.plos.org/plosone/s/figures

Response: We have replaced the supplemental figures S2 and S5 with higher-resolution images.

Reviewer #1

Very interesting in-depth experiment on a novel group of GATA4 inhibitors. Comparison of the effectiveness of several compounds reinforces the utility of the most potent variant.

The prognosis of the hepatoblastoma cell line used in the experiment [embryonal] represents ~20% of all hepatoblastomas, with ~60% survival at 24 months. It would be of therapeutic interest to study the effectiveness of these inhibitors on hepatoblastomas with some of the other histologic types, e.g. pure fetal [~30% of all hepatoblastomas with a 24-month survival of ~90%], or small cell undifferentiated [~3% of all hepatoblastomas with a 24-month survival of 0%].

Response: Thank you for this important proposal. We acknowledge that the cell lines used in the study represented only embryonal histology. Therefore, we have now performed additional experiments with two cell lines that are fetal origin (HB-295 and HB-303). The cells were treated with increasing concentrations of 3i-2012 compound for 24 or 48 h. Both cell lines showed response to the compound already at the relatively low concentrations. These results have now been incorporated into the revised manuscript (Figure 2f-g; Results: page 4, lines 144-147; Discussion: page 9, lines 307-309).

Excellent graphs and pictures.

Some typos:

Line 47: ... effects of a novel ...;

Line 61: ... occur at any age of ...;

Line 85: 'CETSA", not "CESTA".

Line 134: ... tested a novel ...;

Line 139: ... compounds in the 3i-2010 ...;

Line 140: ... viability at the highest ...;

Line 141: ... already at a concentration ...;

Line 169: ... compounds' influence ...;

Line 179: ... at a concentration ...;

Line 179: ... spheroid viability by 85% ...;

Line 244: ... Consistent with the ...;

Line 289: ... survival may be mediated ...;

Line 296: ... to 3i-2012 in upcoming studies ...;

Line 315: ... are arrested at the G2/M phase.;

Line 319: ... at least part of the actions ...;

Line 322: ... have different histologic and genetic ...;

Line 330: Furthermore, a notable ...;

Line 331: ... response to 3i-2012 is regulated ... [proportion ... is ...];

Line 334: The synthesis of ...;

Line 452: ... were performed using IBM ...;

Line 462: ... by the Jane ...; ... the Sigfrid ...;

Line 463: ... the Helsinki ....

Response: We thank the Reviewer for pointing these out. The typos have now been fixed in the revised manuscript.

Reviewer #2

In the manuscript entitled "GATA4-targeted compounds induce apoptosis and diminish viability of hepatoblastoma cells," the authors screened compounds targeting GATA4 to provide potential new drug candidates for treating diseases such as hepatoblastoma. This work is valuable; however, the following issues should be addressed before the manuscript can be considered for further processing.

The authors performed initial screening and identified compounds exhibiting cytotoxicity. Potential related mechanisms were investigated via high-throughput screening. Nevertheless, it remains unclear whether the selected compounds are effective in vivo.

Response: The Reviewer points out an important issue. We fully agree that evaluating the efficacy of the compounds in an animal model would be highly valuable. However those investigations are very time-consuming and laborious (e.g. synthesis of large amount of compounds, pharmacokinetic, and toxicological studies). Therefore, we consider this an important direction for our future studies. This issue has now been discussed in the revised manuscript (Discussion: page 10, Lines 333-335).

Additionally, the direct target proteins of these compounds have not been explored. This study would be significantly strengthened if the pharmacological and efficacy investigations were conducted in greater depth and supported by more comprehensive evidence.

Response: We thank the Reviewer for this important comment. We agree that identifying direct molecular targets and further deepening the pharmacological characterization would strengthen the study. In order to investigate the potential off-targets of 3i-2012 compound, a commercial KINOMEscan screening was utilized. This assay revealed inhibition of a defined subset of kinases (FLT3, KIT, CSF1R, PDGFRA/B, AURKB, and TRKA). The RNA sequencing of 3i-2012 treated hepatoblastoma cells showed that of these potential alternative targets, only PDGFRB and AURKB were differentially expressed. PDGFRA remained unaltered and the expression of FLT3, KIT, and TRKA was negligible both in control and 3i-2012 treated hepatoblastoma cells (Results: page 7, lines 225-232). These results suggest that PDGFRB and AURKB may be among the potential off-target proteins affected by 3i-2012, however, additional studies are needed to confirm this.

We fully acknowledge the Reviewer’s point that comprehensive target deconvolution such as biochemical binding assays in hepatoblastoma cells or genome-wide CRISPR-based target dependency analyses would provide additional mechanistic certainty. Such work is substantial and is planned for our future studies aimed at preclinical development of these compounds. This has now been further discussed in the revised manuscript (Discussion: page 9, lines 296-299).

---

## [Decision Letter · Decision Letter 1]

27 Jan 2026

GATA4-targeted compounds induce apoptosis and diminish viability of hepatoblastoma cells

PONE-D-25-12058R1

Dear Dr. Pihlajoki,

We’re pleased to inform you that your manuscript has been judged scientifically suitable for publication and will be formally accepted for publication once it meets all outstanding technical requirements.

Kind regards,

Consolato M. Sergi

Academic Editor

PLOS One

Additional Editor Comments (optional):

Please address the requests of the reviewers thoroughly.
---

## [Editor Report · Acceptance letter]

PONE-D-25-12058R1

PLOS One

Dear Dr. Pihlajoki,

I'm pleased to inform you that your manuscript has been deemed suitable for publication in PLOS One. Congratulations! Your manuscript is now being handed over to our production team.

Kind regards,

on behalf of

Professor Consolato M. Sergi

Academic Editor

PLOS One